# Effect of the Gestational Fluoxetine Administration on Behavioral Tests and Hippocampal Structure in Male Offspring of Rats

**DOI:** 10.3390/ijms262110758

**Published:** 2025-11-05

**Authors:** Marcelo Gustavo Lopes, Gabriel Boer Grigoletti-Lima, Patrícia Aline Boer, José Antonio Rocha Gontijo

**Affiliations:** Fetal Programming and Hydroelectrolyte Metabolism Laboratory, Medicine and Experimental Surgery Center, Internal Medicine Department, Faculty of Medical Sciences, State University of Campinas, Campinas 13083-970, SP, Brazil; marcelolopes46@gmail.com (M.G.L.); gabrielboergrigoletti@gmail.com (G.B.G.-L.); alineboer@yahoo.com.br (P.A.B.)

**Keywords:** fluoxetine, hippocampal cellularity, neurogenesis, fetal development, gestational behavior

## Abstract

Depression is a common mental disorder during gestation, posing potential risks to fetal development and leading to behavioral and psychiatric alterations in offspring. Pharmacological intervention, particularly with selective serotonin reuptake inhibitors (SSRIs), is often necessary. This study investigated the effects of fluoxetine (F) on behavioral and memory changes in rodent offspring following maternal gestational and lactation treatment, as well as potential alterations in hippocampal cellularity compared to control (C) progeny. Methodologies included the Morris water maze, elevated plus maze, activity monitoring, parental behavior assessments, and isotropic fractionation for the quantification of hippocampal cells and neurons. Results indicated that maternal fluoxetine exposure significantly affected the body mass, brain weight, and hippocampal metrics of the offspring, aligning with the ‘selfish brain’ hypothesis. Notably, dams treated with fluoxetine showed reduced parental care, leading to offspring with increased activity levels but no changes in anxiety-like behaviors. However, while there was a decline in learning and memory retention, as assessed by the Morris water maze, working and reference memory did not differ significantly from those of controls. This study establishes an association between fluoxetine treatment, increased hippocampal neuron density, and behavioral changes related to memory and hyperactivity, with implications for understanding behavioral disorders and informing future therapeutic interventions.

## 1. Introduction

Previous studies in our laboratory have shown that factors such as gestational malnutrition and tobacco use negatively impact the structural and neurotransmitter development of central nervous system regions involved in behavioral regulation in rodent models [1,2,3,4,5,6,7]. Depression is a common mental disorder that can occur during and outside of pregnancy. The effective management of depression is advised during pregnancy and immediately postpartum to reduce adverse effects on both the mother and her offspring [8,9]. Approximately 10% of women experience major depressive disorders during pregnancy or postpartum, often due to hormonal changes, contraceptive use, and pregnancy-related challenges [8,10]. Maternal depression can lead to attention deficits and behavioral issues in offspring during adolescence and adulthood [11,12,13]. Pharmacological intervention is necessary for about 2–3% of pregnant women, with selective serotonin reuptake inhibitors (SSRIs) being the most commonly prescribed due to their perceived safety and mild side effects [14,15]. However, there is limited information regarding the long-term neurodevelopmental outcomes associated with prenatal exposure to SSRIs [16]. Research on prenatal fluoxetine treatment has produced mixed findings, particularly concerning potential fetal abnormalities. Serotonin (5-HT) plays a crucial role in brain and organ development [17,18]. During gestation, serotonin receptors are essential for axonal and dendritic differentiation [19,20] and influence cellular processes during fetal and early postnatal development [21,22]. Variations in tryptophan metabolism, influenced by genetic, epigenetic, or environmental factors, may alter fetal development [23]. Furthermore, SSRIs can cross the placenta and affect fetal serotonergic modulation [24,25]. Studies indicate that fluoxetine concentrations in umbilical cord blood closely resemble those in maternal serum [26]. Research has confirmed that selective serotonin reuptake inhibitors (SSRIs) can be transmitted through breast milk, with levels comparable to those in maternal serum (Hendrick et al., 2001) [27]. Serotonergic receptors interact with serotonin (5-HT), which is crucial for brain development and neuronal responses. While SSRIs during pregnancy have not been linked to significant fetal malformations [16,28], they may increase the risk of premature births, intrauterine growth restriction, and behavioral disorders in children [29]. Some studies have suggested associations between maternal SSRI exposure and developmental delays or autism spectrum disorders [30,31,32], while others found no significant cognitive impacts [33]. It is important to recognize that untreated severe depression in pregnancy can lead to fetal complications, as maternal stress is linked to neurodevelopmental issues [34]. The hippocampus, vital for emotion, memory, and spatial learning, consists of the dorsal and ventral regions—with the dorsal focused on spatial memory and the ventral more responsive to stress. Its structure includes subfields like CA1, CA3, and the dentate gyrus (DG) [35]. The subventricular zone (SVZ) contains stem cells that migrate to the hippocampus as interneurons [36]. Research in rodents has advanced our understanding of hippocampal function, highlighting similarities with humans [37]. Severe maternal protein restriction in rats has a negative impact on neurodevelopment, affecting hippocampal morphology and the distribution of neurotransmitter receptors [38]. The current study will investigate behavioral and memory changes in rodent offspring exposed to fluoxetine during the perinatal period, as well as its potential impact on hippocampal neurogenesis.

## 2. Results

### 2.1. Animals and Experimental Groups 

Fetal losses were recorded in seven dams from the control group (*n* = 16) and two dams from the fluoxetine group (*n* = 11), all under the same environmental conditions. Offspring from dams administered oral fluoxetine (group F) exhibited a significantly lower body mass at birth compared to the control group (group C) for both male (C: 6.75 ± 0.06 g, *n* = 41, vs. F: 6.35 ± 0.04 g, *n* = 52; *p* < 0.0001) and female offspring (C: 6.38 ± 0.06 g, *n* = 43, vs. F: 6.04 ± 0.05 g, *n* = 41; *p* < 0.0001) (see Figure 1A,B). Analysis from 21 to 42 days showed consistently reduced body mass in group F at all ages (e.g., 42nd day—C: 207.2 ± 3.046 g, *n* = 27, vs. F: 196.4 ± 3.4 g, *n* = 36; *p* < 0.05) (see Figure 1C). A significant decline in the maternal care index was observed in females treated with fluoxetine, particularly between days 4 and 8 postpartum (see Figure 1D). No significant differences were found in overall brain mass between the groups (C: 0.568 ± 0.009 g, *n* = 10, vs. F: 0.589 ± 0.008 g, *n* = 12; *p* = 0.06), although the fluoxetine group exhibited a significantly higher hippocampal mass at 42 days (C: 0.27 ± 0.028 g, *n* = 5, vs. F: 0.35 ± 0.0068 g, *n* = 5; *p* = 0.02) (see Figure 2B).

### 2.2. Offspring Behavioral Tests

#### 2.2.1. Morris Water Maze

No differences in working memory were found between groups (see Figure 3A). However, from the fourth to seventh days, fluoxetine offspring showed increased latency in finding the hidden platform, which, while not statistically significant, warrants attention (see Figure 3B). On the eighth day, fluoxetine offspring spent less time in the quadrant of the former platform compared to controls (C: 31.5 ± 1.9 s, *n* = 19, vs. F: 24.3 ± 1.6 s, *n* = 22; *p* < 0.05) (see Figure 3C).

#### 2.2.2. Open Field Activity Monitoring Test

No significant differences were noted in distance traveled (C: 9782 ± 583.5 mm, *n* = 19, vs. F: 10,450 ± 374.0 mm, *n* = 22) (see Figure 4A, Table 1). However, fluoxetine offspring demonstrated greater activity (C: 1.234 ± 0.09 min, *n* = 20, vs. F: 1.455 ± 0.05 min, *n* = 22; *p* = 0.0224) (Table 1) and spent more time in the center of the arena (C: 11.11 ± 1.7 s, *n* = 19, vs. F: 14.86 ± 1.3 s, *n* = 21; *p* = 0.0479) (see Figure 4C). Fluoxetine offspring also produced more fecal boluses, indicating elevated fear (F: 4.174 ± 0.5, *n* = 23, vs. C: 1.857 ± 0.4, *n* = 21; *p* = 0.0015) (see Figure 4D).

#### 2.2.3. Elevated Plus Maze Test

The results from the elevated plus maze indicated no behavioral differences between control and fluoxetine offspring (e.g., time in open arms—C: 21.20 ± 3.0 s, *n* = 20, vs. F: 19.63 ± 3.0 s, *n* = 19) (see Figure 5).

### 2.3. Isotropic Fractionation

No statistical differences were found in total hippocampal cell numbers (see Figure 6A). However, the fluoxetine group had an increased number of non-neuronal cells, which, although not statistically significant, contributes to the existing knowledge (see Figure 6C). The number of hippocampal neurons increased by 57% in fluoxetine offspring compared to controls (see Figure 6B).

### 2.4. Immunohistochemistry

The current study demonstrated significant differences in F offspring, which showed a notable increase in stem cell numbers within the granular cell layer and subgranular zone (GCL and SGZ) of the dentate gyrus compared to the control group (see Figure 7A,B). While F offspring exhibited increased mitotic activity, no statistically significant differences were found (refer to Figure 7B). Furthermore, there were no significant differences in the number of mitoses among the stem cells in the subventricular zone (SVZ) of the lateral ventricles between fluoxetine-treated and control offspring (see Figure 7C).

## 3. Discussion

Research has revealed that the use of antidepressants, notably selective serotonin reuptake inhibitors (SSRIs), during pregnancy may adversely affect offspring. Maternal SSRI treatment is linked to an increased risk of developmental lung disorders and reduced birth weight [29]. Additionally, exposure during gestation may cause long-term alterations in the hypothalamic–pituitary–adrenal (HPA) axis; however, the safety of SSRIs during pregnancy remains unclear, complicating decisions about treatment continuation [39,40]. Fluoxetine, one of the most commonly prescribed SSRIs, readily crosses the placental barrier and is transmitted through breastfeeding, potentially affecting fetal development [26,41]. Its prolonged half-life may lead to sustained fetal exposure [42]. Studies indicate that gestational protein restriction results in low birth weight in rodent models, leading to gender-specific changes in blood pressure, glucose metabolism, and anxiety-like behaviors, particularly in males [43,44]. Sex hormones contribute to sexual phenotype dimorphism in the fetal programming model of adult disease by modulating regulatory pathways critical in the long-term control of neural, cardiovascular, and metabolic functions. This study focused exclusively on male rats to minimize the impact of gender-related variables. The current study reinforces previous findings [45] that maternal exposure to fluoxetine during embryonic and fetal development leads to reduced body weight in offspring at birth. Additionally, research indicates that postnatal fluoxetine exposure during breastfeeding, regardless of maternal prenatal stress, significantly lowers the weight of rodent progeny [46,47]. This weight reduction may result from elevated diencephalic serotonin (5-HT) levels, particularly in the hypothalamus, which affects the regulation of hunger and satiety [48]. High levels of serum and central nervous system 5-HT, resulting from selective reuptake inhibition, promote increased satiety and reduced body mass gain [49]. In this study, maternal fluoxetine treatment did not alter the brain and hippocampal mass of female offspring in isolation; however, it increased hippocampal mass when normalized to brain weight. This research also supports the ‘selfish brain’ hypothesis, which suggests that the CNS regulates peripheral energy metabolism to maintain a stable brain energy supply, influencing nutrient allocation and intake. This theory may elucidate the effects of fluoxetine on progeny weight gain. The findings of this study are significant as they highlight potential risks associated with maternal selective serotonin reuptake inhibitor (SSRI) treatment and establish a basis for further research. Although overall brain mass remained unchanged, notable hippocampal cytological modifications were observed in F progeny. Following maternal SSRI treatment, male F offspring exhibited reduced body mass without corresponding changes in brain weight, which was linked to an increase in hippocampal neurons. Furthermore, the reduced birth weight of F offspring may be linked to elevated fetal corticosteroid levels and decreased growth hormone secretion, as previously documented [50]. This reduced body mass persisted throughout the six-week experimental period compared to controls. Additionally, this study demonstrated a decrease in the maternal care index among fluoxetine-treated dams during the initial postpartum days relative to the control group [51]. While few studies have investigated postnatal maternal care following perinatal fluoxetine treatment, existing research presents conflicting results regarding the effects on maternal care behavior [39]. In terms of anxiety-like behavior, the elevated plus maze test revealed no significant differences between offspring from control and fluoxetine-treated progeny, as indicated by similar entries in the open and closed arms [51,52,53]. Previous studies are contradictory; while McAllister et al. (2012) and Kiryanova et al. (2016) described a reduced level of anxiety in gestational SSRI-treated offspring [54,55,56], others did not show any changes. The current study’s open-field analysis revealed increased overall activity in the F-offspring compared to the control progeny, although average speed and total distance covered did not differ significantly between the groups. The F-treated offspring spent more time in the central area of the open field apparatus, suggesting reduced anxiety-related emotionality. However, the findings also indicated increased fecal bolus production, reflecting a heightened fear state in the F offspring. Previous studies by Kiryanova et al. [55,56] examined the effects of prenatal stress in rodents treated with fluoxetine, revealing hyperactivity in progeny exposed to stress without SSRI treatment. It is worth noting that Kiryanova et al. [56] focused on time and distance traveled, which showed no change in the present study. Additionally, Noorlander et al. (2008) found that F offspring spent less time in the center of the open field than the control progeny, which contrasts with the current results [51]. The Morris water maze study revealed a reduced capacity for learning and memory retention in F offspring following the removal of the hidden platform, compared to the control group. However, assessments of working memory, linked to the prefrontal cortex, and reference memory, associated with the hippocampus, did not show significant differences between the two groups. This indicates that the effects of selective serotonin reuptake inhibitor (SSRI) treatment on offspring behavior and memory retention are still not fully understood. Previous studies found no significant differences in learning and memory in rodents treated with SSRIs compared to controls, primarily because they did not use the hidden platform strategy for memory assessment [28,54]. In this study, F progeny from SSRI-treated dams exhibited increased motor activity and fear behaviors in the open-field test, correlating with diminished learning and memory as measured by the Morris water maze. If confirmed through further research, these findings may greatly inform our understanding of SSRIs’ impact on offspring behavior and memory retention, particularly concerning the treatment of attention deficit hyperactivity disorder (ADHD) [57]. To our knowledge, no prior research has linked SSRI usage to the development of ADHD in offspring rodents. Previous retrospective clinical studies have suggested an increased risk of autism spectrum disorders (ASD) in children exposed to antidepressants during the prenatal period. The prevalence of ASD and attention-deficit/hyperactivity disorder (ADHD) among children in the United States, particularly a combined incidence of nearly 30% in New England, underscores the need for further investigation. Our multiple logistic regression analyses have shown an association between prenatal antidepressant exposure and ASD risk in 1377 affected children and in 2243 diagnosed with ADHD. However, adjusting for sociodemographic factors revealed no significant relationship between antidepressant exposure during pregnancy and ASD risk. It is essential to consider the modest risk of ADHD and ASD resulting from prenatal exposure against the substantial consequences of untreated maternal depression. In another study [58] involving approximately 38,000 children, fewer than 500 were diagnosed with ADHD, and there was no link between prior SSRI use and the condition, with a notable association with maternal psychiatric illness instead. Conversely, studies have found a direct association between gestational exposure to SSRI-class antidepressants and increased ADHD risk, excluding variations in maternal psychiatric illness [59,60]. Furthermore, studies indicate that perinatal exposure to fluoxetine modifies neural network architecture in key areas of the hippocampus, potentially leading to behavioral abnormalities [61,62]. However, our current study, which utilized the low-cost, elevated plus maze (LCE) and open field tests, did not indicate increased anxiety-like behavior in offspring of dams treated with fluoxetine during pregnancy and lactation [53] but did demonstrate significant changes in memory and learning in these offspring. Recent studies have revealed that the regulation of activity and exploration capacities in rodents is not solely influenced by postpartum maternal care. The interaction between hippocampal structural changes and reduced modulation of the prefrontal cortex—responsible for activity, exploration, and executive functions—along with the anterior cingulate gyrus and the amygdala, which are associated with attention, emotional responses, fear, and learning, may explain the findings of this research. Consequently, it can be hypothesized that behavioral changes are linked not only to modifications in maternal care but also to gestational exposure to selective serotonin reuptake inhibitors (SSRIs) [54]; understanding this interaction could aid in developing appropriate therapeutic options for both mothers and their offspring [40,63,64,65]. Evidence from human and animal studies suggests that serotonin and its receptors play a crucial role in modulating the interactions between genetic and environmental factors during brain development, influencing hippocampal brain-derived neurotrophic factor (BDNF) levels. Notably, gestational exposure to fluoxetine has been shown to enhance hippocampal neurogenesis in offspring subjected to prenatal stress, thereby reversing the typical decrease in neurogenesis and highlighting fluoxetine’s neurogenic potential [66]. Our findings indicate that while absolute brain and hippocampal weights are similar, the F progeny displayed significantly greater numbers of hippocampal neurons and primordial cells in the dentate gyrus compared to age-matched controls [29]. This change in cellular profile during the postnatal period in F offspring may correlate with improved learning and memory, particularly in spatial tasks. Additionally, recent findings suggest a connection between memory consolidation and a dynamic cycle of apoptosis and cellular differentiation, with apoptosis predominating in these processes. The reduction in neurons in the hippocampus typically occurs under various conditions. Data suggest that effective spatial learning relies primarily on the programmed death and reduction in newly differentiated neurons in this brain region. Hence, examining the impact of selective serotonin reuptake inhibitors (SSRIs) on memory retention, measured by the Morris water maze, may indicate that behavioral changes are linked to decreased neuronal cell death in rodent hippocampi. Previous studies have shown that learning improves with a decline in newly differentiated neurons during specific stages of fetal development. This study hypothesizes that inhibiting programmed cell death—where cells are destined to die—could result in an increased number of neurons, impairing memory retention and raising activity levels in fluoxetine-treated animals. These findings suggest that neuronal networks are shaped through a regulated mechanism involving the survival or death of neurons, which may be disrupted by fluoxetine treatment. Additionally, maternal stress during pregnancy has been associated with a decrease in hippocampal neurogenesis. Previous study [66] indicated that fluoxetine exposure could reverse the decline in neurogenesis in the offspring of stressed mothers. Notably, fluoxetine administration during gestation and lactation resulted in a significant reduction in body mass during follow-up, without an increase in brain mass, particularly in the hippocampus, suggesting a potential protective mechanism [67]. While there were no differences in total hippocampal cell counts across the experimental groups, the offspring of mothers treated with fluoxetine displayed an increase in neuron numbers linked to higher stem cell counts in the dentate gyrus. Additionally, maternal care decreased among fluoxetine-treated dams compared to controls. The Morris water maze results revealed significant changes on the eighth day, indicating reduced learning in the offspring of mothers treated with fluoxetine. This study highlights the relationship between fluoxetine treatment, elevated hippocampal neuron counts, and changes in memory retention and hyperactivity. Further research is needed to explore the implications of these findings regarding behavioral disorders and related neurodevelopmental factors.

## 4. Materials and Methods

### 4.1. Animals and Experimental Groups 

Animal and experimental group experiments were conducted on age-matched sibling-mated Wistar HanUnib rats (250–300 grams), following guidelines from the Brazilian College of Animal Experimentation (COBEA), and were approved by the Institutional Ethics Committee (CEUA/UNICAMP #4571-1/2017). The local colony was derived from breeding stock supplied by CEMIB/UNICAMP, Campinas, SP, Brazil. After weaning at three weeks, the rats were kept in controlled conditions (23 ± 2 °C and a 12-hour light/dark cycle) with ad libitum access to water and standard chow. From the 8th to the 11th week, female rats were given a small piece of nut-flavored wafer cookies soaked in saline daily for adaptation. At 11 weeks, they were mated; sperm presence in the vaginal smear marked day 1 of pregnancy. Females were divided into a control (C) and a fluoxetine administration (F) group, with the F group receiving fluoxetine hydrochloride (10 mg/kg/day) in nut-flavored wafer cookies, while the C group received saline. This regimen continued during gestation and breastfeeding (21 days). Dams were housed in individual cages and monitored for maternal behavior care, and litter sizes were adjusted to eight male pups per mother. In the present study, all male or female animals after birth were used to measure birth weight; however, only the male pups were studied. After weaning, control (*n* = 40) and fluoxetine (*n* = 40) groups were established. 

### 4.2. Experimental Procedures—Behavioral Analysis

#### Maternal Maternal Care Behavior

Maternal behavior was assessed daily from birth until the end of weaning (21 days post-birth) at 10:00 AM. Dams were kept with their offspring in individual cages, with minimal handling. Behavioral assessments were based on established scoring criteria (Table 2).

### 4.3. Offspring Behavioral Tests

#### 4.3.1. Morris Water Maze (MWM) 

The Morris water maze test is a standard spatial learning task in rats, conducted in a circular black tank (170 cm diameter, 31 cm deep, 22 °C) placed in a dimly lit area with external cues. The tank was divided into quadrants, with a black platform positioned in one quadrant.

#### 4.3.2. Working Memory Task 

This task evaluated prefrontal cortex (PFC) functionality, assessing rats’ ability to learn the position of a hidden platform across four consecutive trials over four days. The platform varied daily yet remained constant each day. Rats started at one of four points (N, E, S, or W) and concluded trials upon accessing the platform. If unsuccessful after 120 s, they were guided to the platform and escape latency was recorded. Rats could remain on the platform for 30 s prior to repositioning.

#### 4.3.3. Reference Memory Task 

The reference memory task assessed hippocampal function, with rats learning the platform’s position over four days of acquisition (four trials per day). The platform remained in a constant quadrant, and the same methodology was applied for trial completions as in the working memory task. Escape latency was recorded throughout the trials.

#### 4.3.4. Activity Monitoring Test

In the sixth week of life, male offspring from each litter were categorized into control (C) and fluoxetine (F) subgroups and underwent a five-minute assessment using an activity monitor (plate number 1, Insight, Brazil). This monitor featured six bars with 16 infrared sensors to track the subject’s position in an acrylic enclosure measuring 500 mm × 480 mm × 500 mm. The subjects were placed in the center, and their movements were recorded. The results include the average number of times behaviors occurred, such as rearing, dislocation, freezing, self-grooming, and crossing. Student’s *t*-test was also used for comparison. Additionally, the percentage of fecal boluses, indicative of fear response, was noted after each session.

#### 4.3.5. Elevated Plus Maze

At six weeks after birth, male offspring from the C and F subgroups were assessed for five minutes in an elevated plus maze using software for sensor monitoring (document number 1, Insight, Brazil). Animals were introduced into the maze’s center, facing the closed arms. The time spent in open arms versus closed arms indicated anxiety levels. Metrics included the percentage of open-arm entries and the percentage of time spent in open arms. Total entries were analyzed to evaluate general motor activity.

### 4.4. Hippocampal Total Cell and Neuron Quantification by Isotropic Fractionation

Cell and neuron quantification followed methods discussed in sources [7,68]. Six-week-old control (C) and age-matched fluoxetine (F) offspring (*n* = 5 per group) were anesthetized with isoflurane and perfused transcardially with saline and a 4% paraformaldehyde solution. The hippocampus was mechanically dissociated and homogenized in a solution of sodium citrate and Triton X-100. Nuclei were resuspended in a DAPI solution, and nuclear density was assessed using a hemocytometer. Total cell counts were estimated based on nuclear density. Neurons were identified by incubating a portion of the nuclear suspension with an anti-NeuN antibody, followed by a Cy3-conjugated secondary antibody for counting under a fluorescence microscope.

### 4.5. Immunohistochemistry

Immunohistochemical analysis was performed on six-week-old control (C) and fluoxetine-treated (F) offspring (*n* = 5). Animals were anesthetized and underwent tissue perfusion with a heparinized saline solution and a 4% paraformaldehyde solution. Following immersion fixation, the brains were embedded in paraplast, and 5 µm sections were cut. After deparaffinization and antigen retrieval, sections were incubated overnight at 4 °C with primary antibodies: SOX2 and Ki-67. Secondary antibodies conjugated with fluorochromes were applied, and slides were mounted using Vectashield containing DAPI. Image analysis was conducted using an Olympus BX51 Fluorescence Microscope, Melville, NY, USA.

### 4.6. Data Presentation and Statistical Analysis

Cognitive performance assessments and structural evaluations were analyzed in a blinded manner by trained observers. Data are presented as mean ± SEM. Time-dependent data were examined using appropriate statistical methods, including ANOVA or Kruskal–Wallis one-way analysis of variance. Bonferroni’s post hoc test was employed for significant ANOVA results. Comparisons of two independent samples were conducted using Student’s *t*-test or the Mann–Whitney U test. A *p*-value of less than 0.05 indicated statistical significance. Data analysis was completed using GraphPad Prism 5.00, Boston, MA, USA.

## Figures and Tables

**Figure 1 ijms-26-10758-f001:**
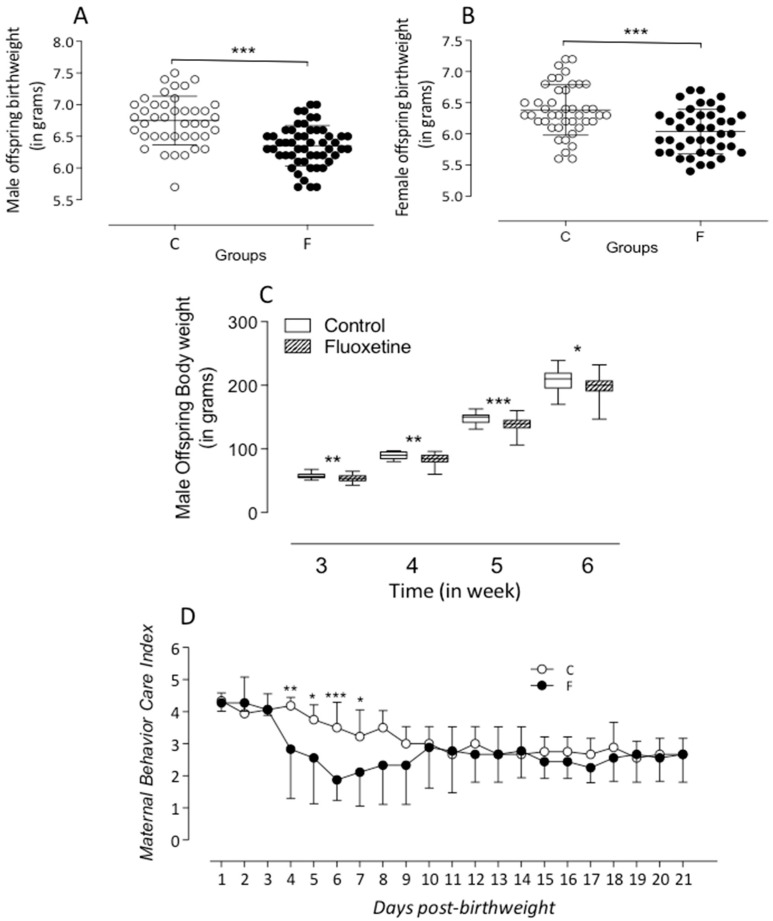
The figure shows the body weight (g) at birth of male (**A**) and female (**B**) fluoxetine (F) offspring. In the present study, after birth, all male or female animals were used to measure birth weight; however, only the male pups were studied and after lactation to 42 days of age (**C**), as compared to control (**C**) (offspring). The figure also includes the representative evolution of the maternal behavior care index (**D**) in the C and F groups during breastfeeding days. Our results, expressed as means ± SEM, were obtained over time and were analyzed using repeated-measures ANOVA. Fluoxetine exposure during pregnancy and breastfeeding significantly affected the offspring’s body weight and maternal care. The significance level was set at * *p* < 0.05, ** *p* < 0.001, and *** *p* < 0.0005.

**Figure 2 ijms-26-10758-f002:**
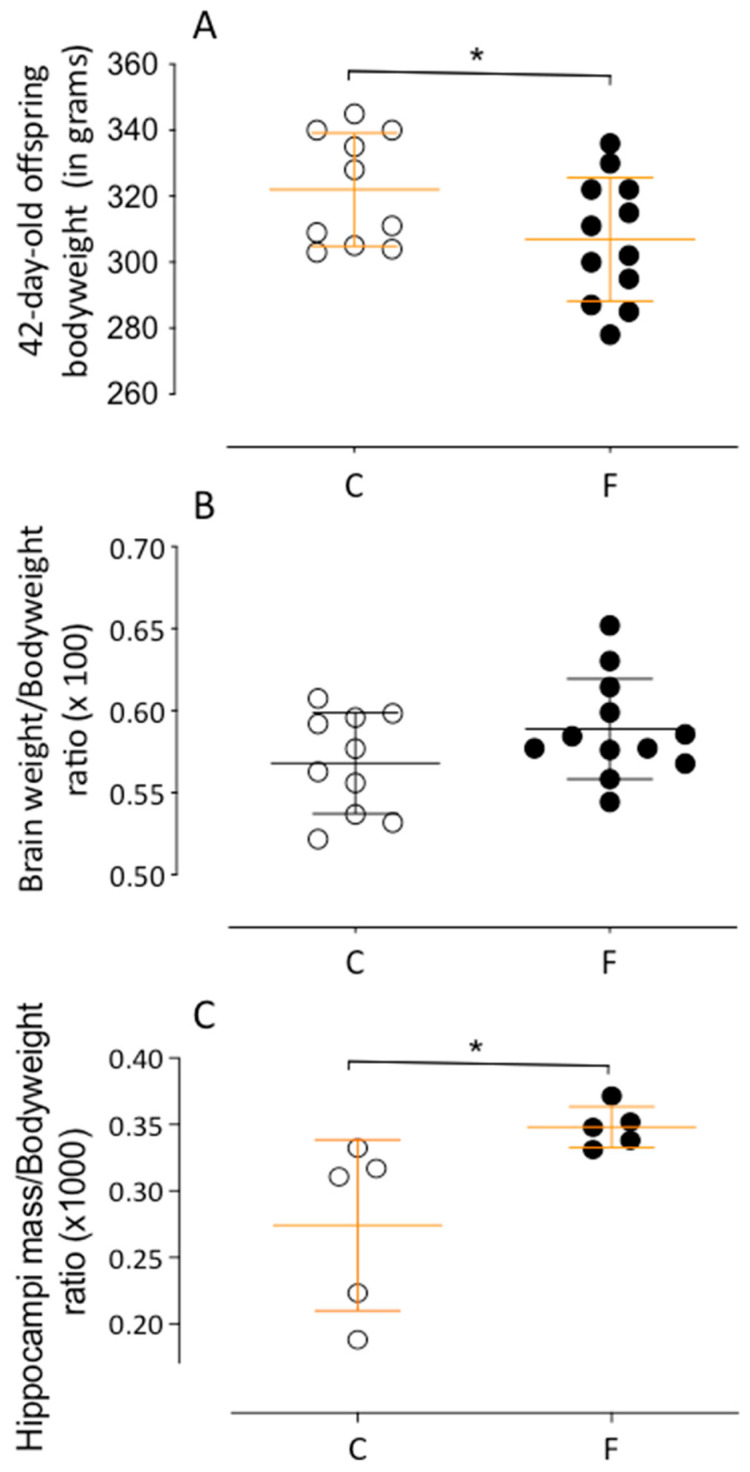
Representative graphics of the hippocampal effects of maternal and breastfeeding fluoxetine treatment on 42-day-old male F offspring compared to age-matched male C offspring on body weight, in grams (**A**), brain weight by body weight ratio (×100) (**B**), and hippocampal mass by body weight (×1000) (**C**). Results are depicted as a scatter dot-plot and are expressed as means ± SEM; comparisons involving only two means within or between groups were performed using Student’s *t*-test. The level of significance was set at * *p* < 0.05.

**Figure 3 ijms-26-10758-f003:**
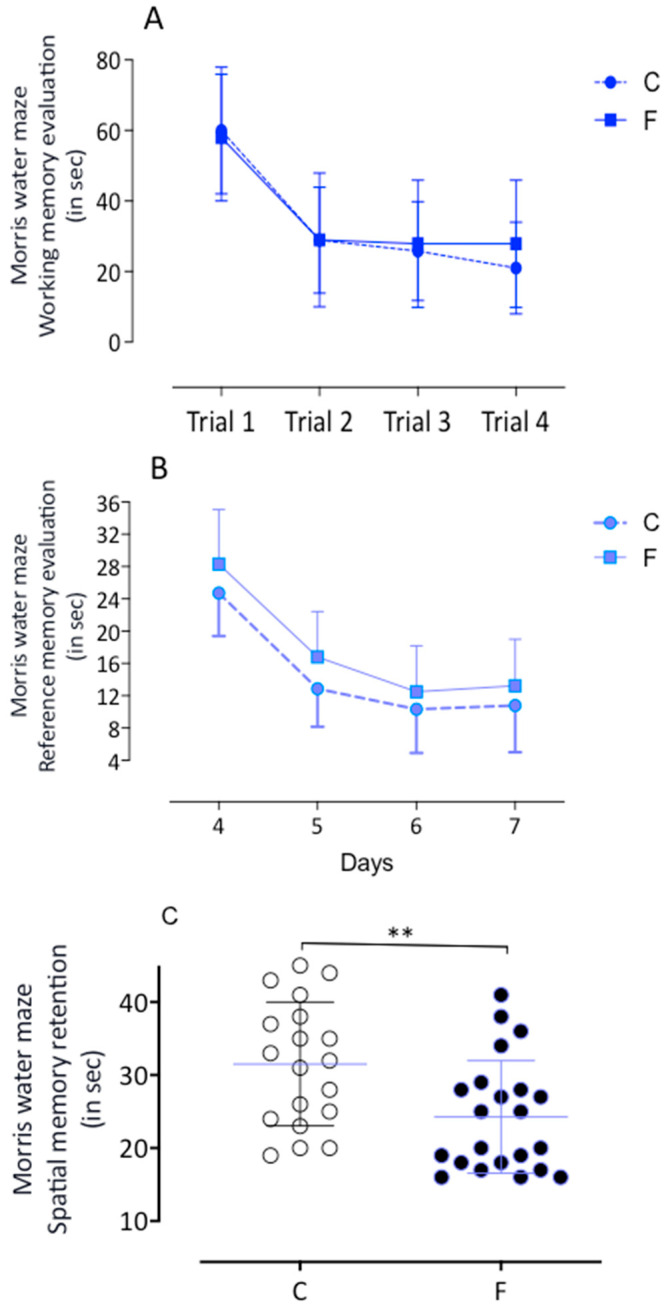
The Morris water maze training performance is illustrated (panels (**A**–**C**)). In the working memory test, the F animals demonstrated a similar learning rate to the C offspring. As observed in panel (**B**), there was no change in the reference memory test results in the male F progeny when compared to the male control (**C**) group. However, the most significant finding is the memory retention results, where on the eighth day, the F offspring remained significantly longer in the quadrant where the platform was on the previous day than the C progeny rats, suggesting superior memory retention ability. This finding has significant implications for the impact of fluoxetine exposure on memory retention. ** Indicates *p* < 0.05.

**Figure 4 ijms-26-10758-f004:**
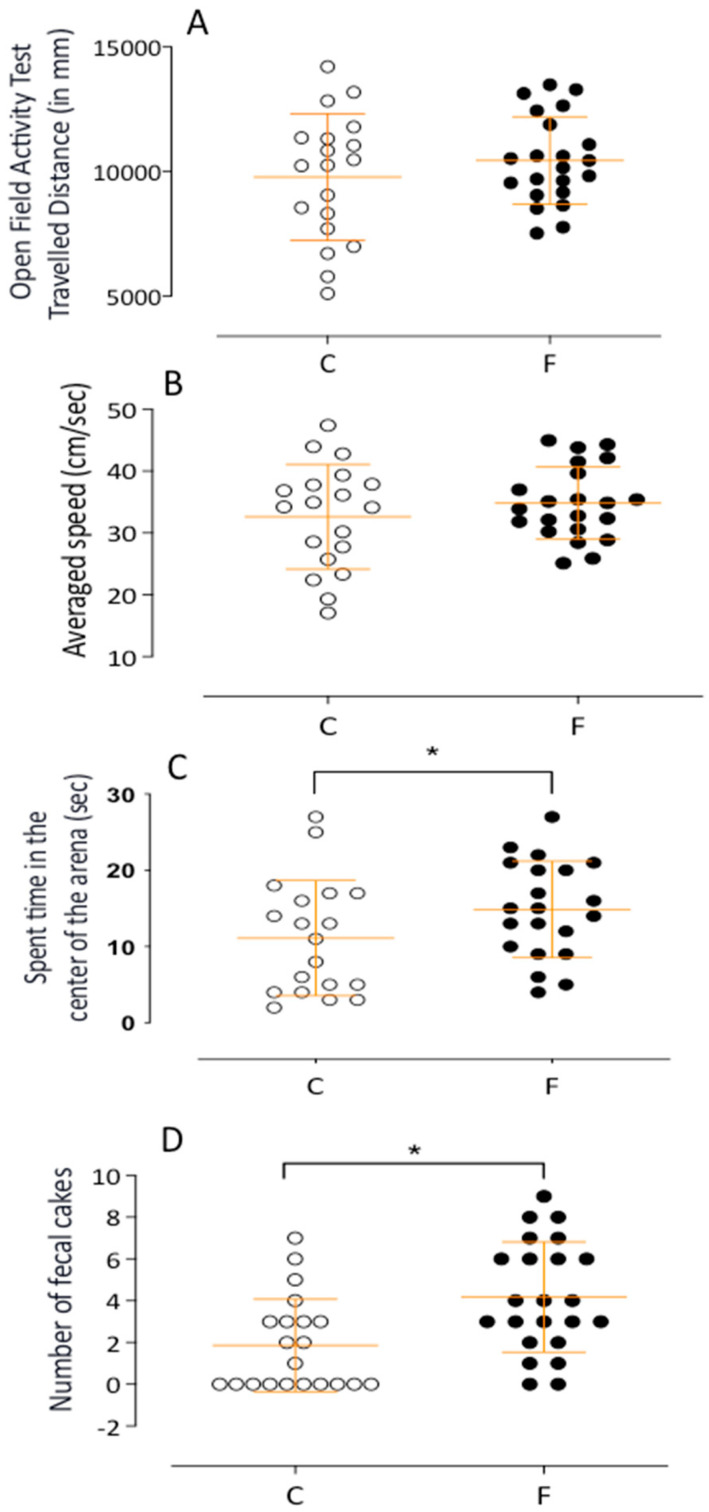
The figure presents a dot plot representing results obtained by the activity monitor in the open field test and the level of activities. In the (**A**,**B**) panels, we demonstrate the traveled distance (in mm) and average speed (in cm/sec) during the open field test, and in the (**C**,**D**) panels, we show the time spent in the center of the arena and the fecal boluses as means ± SEM for male F and male C offspring. The most significant result is that the offspring from mothers treated with fluoxetine exhibited a significantly enhanced number of fecal pellets and spent time in the center of the arena. This finding underscores the impact of fluoxetine exposure on the offspring’s behavior. Comparisons involving only two means within or between groups were performed using Student’s *t*-test. Welch’s *t*-test was used to correct situations characterized by heteroscedasticity. When statistically significant differences were indicated between selected means, post hoc comparisons were performed using Bonferroni’s contrast test. The significance level was set at * *p* < 0.05.

**Figure 5 ijms-26-10758-f005:**
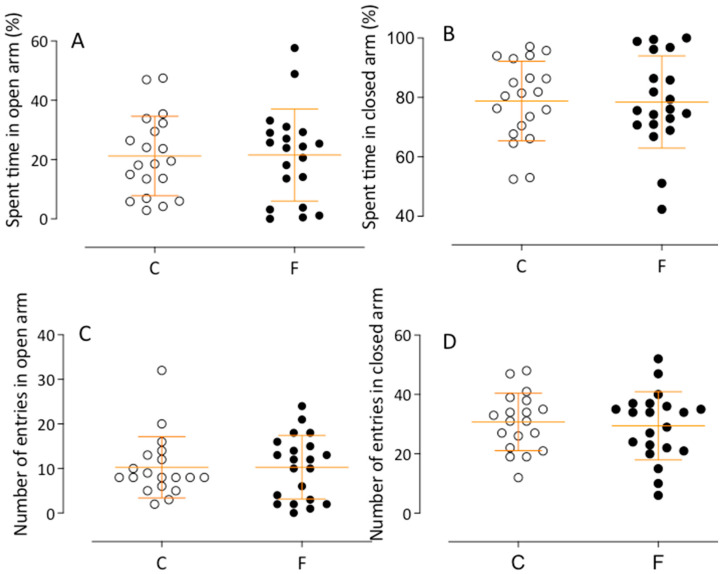
The mean percentage of open and closed arms spent time (**A**,**B**) and open and closed arms entries (**C**,**D**) for offspring in the elevated plus maze is depicted. The exploration in the elevated plus maze was meticulously measured. No significant difference in the time spent and entries into the open arms between male C and male F offspring were observed. Comparisons involving only two means within or between groups were performed using Student’s *t*-test. Welch’s *t*-test was used to correct situations characterized by heteroscedasticity. When statistically significant differences were indicated between selected means, post hoc comparisons were performed using Bonferroni’s contrast test.

**Figure 6 ijms-26-10758-f006:**
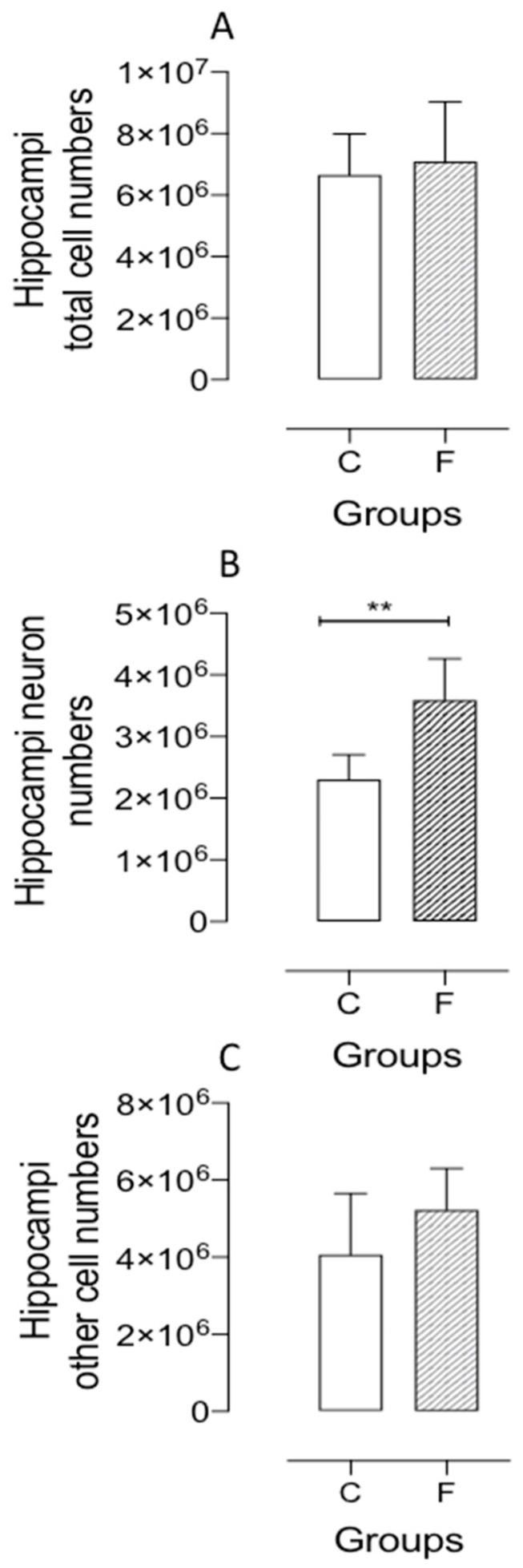
Representative graphics show the hippocampus total cell number (**A** panel); the number of hippocampal neurons (**B** panel) and non-neuronal cells (**C** panel) obtained by the isotropic fractionation technique in the male F offspring compared to the age-matched control male (**C**) group. Results are expressed as means ± SEM; the significance level was set at ** *p* < 0.01. Comparisons involving only two means within or between groups were performed using Student’s *t*-test for each group.

**Figure 7 ijms-26-10758-f007:**
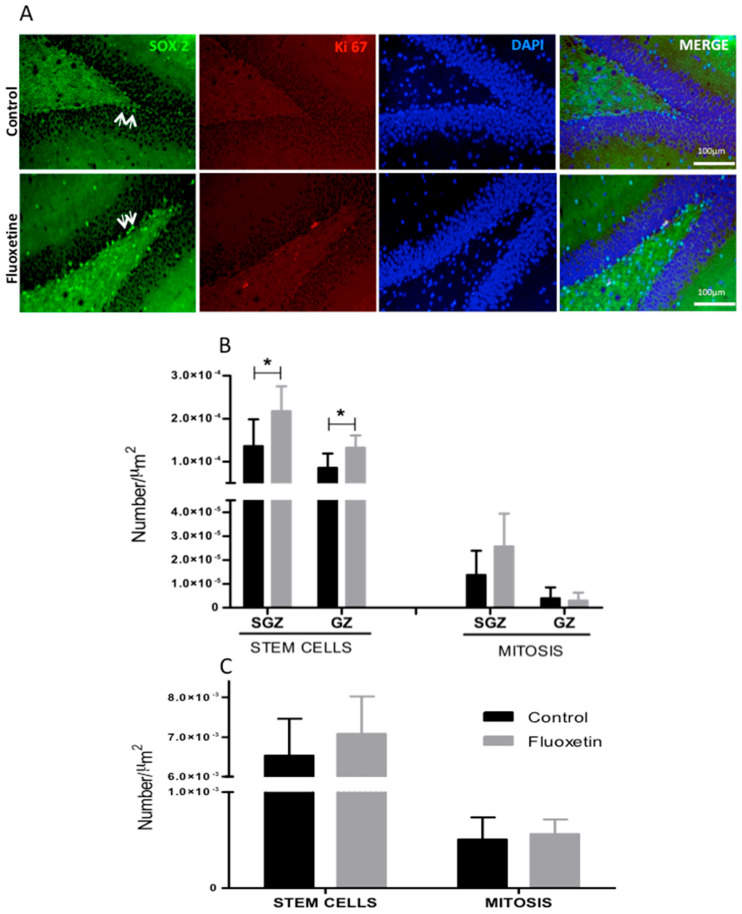
Representative immunohistochemistry: (**A**) Illustration of the number of mitoses (KI-67), stem cells (Sox-2) (**B**,**C** panel), and stem cell mitoses quantified in the subgranular zone (SGZ) and granular cell layer (GCL) of the hippocampal dentate gyrus (GD) in male offspring. On the left side of the panels, the groups of offspring are shown; their mothers were subjected to or not subjected to (control, C) fluoxetine (F) treatment during pregnancy and breastfeeding. Results are expressed as means ± SEM; the significance level was set at * *p* < 0.05 (one-way ANOVA analysis indicated statistical differences between groups; Bonferroni’s contrast test was performed for post hoc comparisons between means for each group.

**Table 1 ijms-26-10758-t001:** Shows result from an open field test of 42-day-old male offspring from the control (C) and fluoxetine (F) groups. The results include the average number of times behaviors occurred, such as rearing, dislocation, freezing, self-grooming, and crossing. We used Bonferroni’s test to check for differences between the groups, considering results significant at * *p* < 0.05. Student’s *t*-test was also used for comparison.

Parameters	Control (C, *n* = 20)	Fluoxetine (F, *n* = 22)	*p* Values
Orthostatic position (frequency)	42.65 ± 3.215	44.55 ± 1.969	0.3053
Activity time (min)	1.234 ± 0.09	1.455 ± 0.056	0.0198 *
Travelled distance (mm)	9478 ± 631.2	10,450 ± 374	0.0926
Speed (cm/sec)	36.6 ± 8.48	34.82 ± 5.84	0.82
Time in arena center (sec)	11.11 ± 1.73	14.86 ± 1.38	0.0479 *

**Table 2 ijms-26-10758-t002:** Maternal care behavioral study protocol (modified from Sodersten & Eneroth, 1984 [68].

Score	Observed Parameters
0	Absence of nest
1	Presence of nest
2	All the pups in the nest
3	All the pups in the nest, with their mother
4	All the pups in the nest, being suckled with arched backs
4.5	All the pups in the nest, being nursed in a passive position, on their side or supine
5	All the pups in the nest, being suckled and licked

## Data Availability

Availability of data and material in: http://repositorio.unicamp.br/acervo/detalhe/1081913?guid=1721417496993&returnUrl=%2fresultado%2flistar%3fguid%3d1721417496993%26quantidadePaginas%3d1%26codigoRegistro%3d1081913%231081913&i=4 (accessed on 1 December 2024), and doi: https://doi.org/10.47749/T/UNICAMP.2019.1081913.

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
