# Peer review of "Effect of the Gestational Fluoxetine Administration on Behavioral Tests and Hippocampal Structure in Male Offspring of Rats"

_ijms, 2025, doi:10.3390/ijms262110758_

Round 1

Reviewer 1 Report

Comments and Suggestions for Authors

This research examines behavioral and neuroanatomical alterations following offsprings' exposure to fluoxetine. However, one of my major concerns with this work is that it does not address whether the observed changes are due to the fluoxetine exposure or changes in maternal care, which could be done by rearing control pups with a fluoxetine mother and vice versa. 

Additionally, the methods and title suggest that the experiments are done in male offspring only, but then Fig. 1A shows females. Could the others please clarify whether all other experiments were performed with males only or both males and females?

The abstract states that they found a reduction in anxiety-like behaviors in male offspring but then there is no supporting evidence for this since there are no changes in the elevated plus maze.

It is unclear how the analysis in Figure 6 was conducted. Why aren't neurons and non-neuronal cells presented as a percentage of total cells. How are non-neuronal cells identified?

Please show the images corresponding to the analysis in Fig. 7. 

It is concerning that so many pups died in the control litters. Could this be a reflection of stress that is obscuring the study's findings?

Comments on the Quality of English Language

There are several grammatical errors and misspellings throughout the text:

By stating that depression occurs in the gestational period, I believe that the authors are referencing the mom, but this isn't clear as written. Instead of saying that depression associates with the gestational period (which refers to the fetus), the authors should state that it is common during pregnancy to clarify that it is a maternal mental illness that may have consequences for the offspring, although they are not the ones diagnosed with depression.

Fluoxetine is misspelled as fluoxetin in the Fig. 7.

Entries is misspelled in Fig. 5.

Statement in 304-305 says that F offspring were compared to F offspring.

Line 268 should say the analysis not the analyze

Statement in lines 26-28 is a run-on. It is unclear what is concerning total brain mass.

Author Response

Response from the Authors

Response to Reviewer Comments

International Journal of Molecular Science

Ref.: ijms-3693676

Title: Effect of Gestational Fluoxetine Administration on Behavioral Tests and Hippocampal Structure in Male Rat Offspring

Authors: Marcelo Gustavo Lopes, Gabriel Boer Grigoletti-Lima , Patricia Aline Boer , José Antonio Rocha Gontijo *

Comments from the Editors and Reviewers:

  1. Response to Reviewer #1

First, thank you very much for your criticism. We appreciate your comments and suggestions greatly. Practically all sections of the manuscript were rewritten entirely, experiments were redone, the number of experiments was increased, and many reviewer suggestions were now included in that new version. The grammatical and typographical errors have been corrected and the text edited. The Introduction, Material, Method, and Discussion sections of the manuscript were revised, shortened, and completely rewritten to include the suggestions and comments of reviewers. 

Reviewer #1: This research examines behavioral and neuroanatomical alterations following offspring's exposure to Fluoxetine. However, one of my primary concerns with this work is that it does not address whether the observed changes are due to fluoxetine exposure or changes in maternal care, which rearing control pups could investigate with a fluoxetine-treated mother and vice versa. 

Additionally, the methods and title suggest that the experiments are done in male offspring only, but then Fig. 1A shows females. Could the others please clarify whether all other experiments were performed with males only or both males and females?

The abstract states that they found a reduction in anxiety-like behaviors in male offspring, but then there is no supporting evidence for this, since there are no changes in the elevated plus maze.

It is unclear how the analysis in Figure 6 was conducted. Why aren't neurons and non-neuronal cells presented as a percentage of total cells? How are non-neuronal cells identified?

Please show the images corresponding to the analysis in Fig. 7. 

It is concerning that so many pups died in the control litters. Could this be a reflection of stress that is obscuring the study's findings?

Comments on the Quality of the English Language

There are several grammatical errors and misspellings throughout the text:

By stating that depression occurs in the gestational period, I believe that the authors are referencing the mom, but this isn't clear as written. Instead of saying that depression associates with the gestational period (which refers to the fetus), the authors should state that it is common during pregnancy to clarify that it is a maternal mental illness that may have consequences for the offspring. However, they are not the ones diagnosed with depression.

Fluoxetine is misspelled as fluoxetin in Fig. 7.

The word " entries is misspelled in Fig. 5.

Statement in 304-305 says that F offspring were compared to F offspring.

Line 268 should read 'the analysis,' not 'analysis.' The statement in lines 26-28 is a run-on. It is unclear what is concerning the total brain mass.

Author Response (Reviewer 1): Practically all sections of the manuscript were rewritten entirely, experiments were redone, the number of experiments has been increased, and many reviewer suggestions were now included in that new version. The grammatical and typographical errors have been corrected and the text edited. The Introduction, Material, Method, and Discussion sections of the manuscript were revised, shortened, and completely rewritten to include the suggestions and comments of reviewers. 

All male or female animals after birth were used to measure birth weight.

Regarding the aspects raised by the reviewer about the animals used in the present study, it is appropriate to clarify that dams were housed in pairs under standard laboratory conditions and had access to food and water ad libitum.

This study aimed to evaluate the effects of Fluoxetine use during pregnancy, without involving experimental models of psychiatric diseases. The dosage of Fluoxetine was consistent with well-established guidelines in the literature for rodents.

The Abstract, Introduction, Materials and Methods, and Discussion sections of the manuscript were revised and completely rewritten to include the suggestions and comments of Reviewer 1. In the present study, only male pups were studied and weighed on their birthday, and eight male puppies were kept per mother. On the 7th and 14th postnatal days, the brain, thymus, and adrenals from male NP and LP offspring were weighed. Here, it is crucial to note that sex hormones influence sexual phenotype dimorphism in the fetal-programmed disease model in adulthood through changes in the long-term regulation of neural, cardiac, and endocrine functions. Thus, the present study was performed in male rats, considering the findings above to eliminate interferences due to gender differences [Kwong et al., 2000; Gillette et al., 2017]. However, additional studies evaluating the functionality of the kidneys, lungs, and liver, as well as aspects related to the behavior of animals subjected to psychological or nutritional stresses of both genders, are being conducted in the laboratory. The results obtained will be discussed in the future.

The study demonstrated a decrease in the maternal care index among fluoxetine-treated dams during the initial postpartum days relative to the control group. While few studies have investigated postnatal maternal care following perinatal fluoxetine treatment, existing research presents conflicting results regarding the effects on maternal care behavior. In terms of anxiety-like behavior, the elevated plus maze test revealed no significant differences between offspring from control and fluoxetine-treated progeny, as indicated by similar entries in the open and closed arms, as correctly expressed throughout the abstract and main text. Previous studies are contradictory; while McAllister et al. (2012) and Kiryanova et al. described a reduced level of anxiety in gestational SSRI-treated offspring, others, however, did not show any changes. The current study's open-field analysis revealed increased overall activity in the F-offspring compared to the control progeny, although average speed and total distance covered did not differ significantly between the groups. The F-treated offspring spent more time in the central area of the open field apparatus, suggesting reduced anxiety-related emotionality. However, the findings also indicated increased fecal bolus production, reflecting a heightened fear state in the F offspring. Previous studies by Kiryanova et al. examined the effects of prenatal stress in rodents treated with Fluoxetine, as well as fluoxetine-induced hyperactivity in progeny exposed to stress without SSRI treatment. It is worth noting that Kiryanova et al. focused on time and distance traveled, which showed no change in the present study. Conversely, Noorlander et al. found that F offspring spent less time in the center of the open field than the control progeny, which contrasts with the current results. The Morris water maze study revealed a reduced capacity for learning and memory retention in F offspring following the removal of the hidden platform, compared to the control group. However, assessments of working memory, linked to the prefrontal cortex, and reference memory, associated with the hippocampus, did not show significant differences between the two groups. This indicates that the effects of selective serotonin reuptake inhibitor (SSRI) treatment on offspring behavior and memory retention are still not fully understood. Previous studies found no significant differences in learning and memory in rodents treated with SSRIs compared to controls, primarily because they did not use the hidden platform strategy for memory assessment. In this study, F progeny from SSRI-treated dams exhibited increased motor activity and fear behaviors in the open-field test, correlating with diminished learning and memory as measured by the Morris water maze. If confirmed through further research, these findings may greatly inform our understanding of SSRIs' impact on offspring behavior and memory retention, particularly concerning the treatment of attention deficit hyperactivity disorder (ADHD). To our knowledge, no prior research has linked SSRI usage to the development of ADHD in offspring rodents. Previous retrospective clinical studies have suggested an increased risk of autism spectrum disorders (ASD) in children exposed to antidepressants during the prenatal period. The prevalence of ASD and attention-deficit/hyperactivity disorder (ADHD) among children in the United States, particularly a combined incidence of nearly 30% in New England, underscores the need for further investigation. Multiple logistic regression analyses have shown an association between prenatal antidepressant exposure and ASD risk in 1,377 affected children and 2,243 diagnosed with ADHD. However, adjusting for socio-demographic factors revealed no significant relationship between antidepressant exposure during pregnancy and ASD risk. It is essential to consider the modest risk of ADHD and ASD resulting from prenatal exposure against the substantial consequences of untreated maternal depression. In another study by Figueroa (2010) involving approximately 38,000 children, fewer than 500 were diagnosed with ADHD, and there was no link between prior SSRI use and the condition, with a notable association with maternal psychiatric illness instead. Conversely, Clements (2015) found a direct association between gestational exposure to SSRI-class antidepressants and increased ADHD risk, excluding variations in maternal psychiatric illness. Furthermore, studies indicate that perinatal exposure to Fluoxetine modulates network architecture in key areas of the hippocampus, potentially leading to behavioral abnormalities. However, our current study, which utilized the low-cost elevated plus maze (LCE) and open field tests, did not indicate increased anxiety-like behavior in offspring of dams treated with Fluoxetine during lactation; however, it did demonstrate significant changes in memory and learning in these offspring. Recent studies have revealed that the regulation of activity and exploration capacities in rodents is not solely influenced by postpartum maternal care. The interaction between hippocampal structural changes and reduced modulation of the prefrontal cortex—responsible for activity, exploration, and executive functions—along with the anterior cingulate gyrus and the amygdala, which are associated with attention, emotional responses, fear, and learning, may explain the findings of this research. Consequently, it can be hypothesized that behavioral changes are linked not only to modifications in maternal care but also to gestational exposure to selective serotonin reuptake inhibitors (SSRIs).

Cell and neuron quantification followed the methods discussed in sources [7,66]. Six-week-old Control (C) and age-matched Fluoxetine (F) offspring (n=5 per group) were anesthetized with isoflurane and perfused transcardially with saline and a 4% paraformaldehyde solution. The hippocampus was mechanically dissociated and homogenized in a solution of sodium citrate and Triton X-100. Nuclei were resuspended in a DAPI solution, and nuclear density was assessed using a hemocytometer. Total cell counts were estimated based on atomic density. Neurons were identified by incubating a portion of the nuclear suspension with an anti-NeuN antibody, followed by a Cy3-conjugated secondary antibody for counting under a fluorescence microscope.

Reviewer 2 Report

Comments and Suggestions for Authors

The authors carried out multiple in vivo tests and in vitro tests using reasonable numbers of maternal rats with their offsprings to demonstrate the effects of fluoxetine on the memory, bodyweight etc. Even the workload is very impressive, there is still a major concern about the design of entire experiment.

Major concerns:

  1. The authors used non-disease maternal rats with treatments of fluoxetine, but in clinic, fluoxetine, as an antidepressant drug, usually prescript to the patients with depression issues, in other words, the normal people never take this medicine with normal mental condition. So, it is ideal if the authors could make depression animal model first and then conduct a series of behavioral tests to either mother rats or their offsprings.
  2. The authors just uses one dose of fluoxetine for mother rats, so what is the evidence of this choice?

Minor comments:

  1. There should be a space between the numbers and units, for example, n=5 should be n = 5. Please correct them throughout the manuscript.
  2. Every figure includes “Figure X” in the bottom of them, they are not necessary to present your results, so they should be removed.
  3. Table 2, the first row is empty, so it should be deleted.
  4. Figure 1, as well as Figure 3, for related results, if you applied student’s t-test, you should clarify them in the corresponding figure legends.
  5. All your figure legends are lack of n numbers.

Author Response

Response from the Authors

Response to Reviewer Comments

International Journal of Molecular Science

Ref.: ijms-3693676

Title: Effect of Gestational Fluoxetine Administration on Behavioral Tests and Hippocampal Structure in Male Rat Offspring

Authors: Marcelo Gustavo Lopes, Gabriel Boer Grigoletti-Lima , Patricia Aline Boer , José Antonio Rocha Gontijo *

Comments from the Editors and Reviewers:

  1. Response to Reviewer #2

First, thank you very much for your criticism. We greatly appreciate your comments and suggestions. Practically all sections of the manuscript were rewritten entirely, experiments were redone, the number of experiments was increased, and many reviewer suggestions were now included in that new version. The grammatical and typographical errors have been corrected and the text edited. The Introduction, Material, Method, and Discussion sections of the manuscript were revised, shortened, and completely rewritten to include the suggestions and comments of reviewers. 

Reviewer #2: This research examines behavioral and neuroanatomical alterations following offspring's exposure to Fluoxetine. However, one of my primary concerns with this work is that it does not address whether the observed changes are due to fluoxetine exposure or changes in maternal care, which rearing control pups could investigate with a fluoxetine-treated mother and vice versa. 

Additionally, the methods and title suggest that the experiments are done in male offspring only, but then Fig. 1A shows females. Could the others please clarify whether all other experiments were performed with males only or both males and females?

The abstract states that they found a reduction in anxiety-like behaviors in male offspring, but then there is no supporting evidence for this, since there are no changes in the elevated plus maze.

It is unclear how the analysis in Figure 6 was conducted. Why are n'tns and non-neuronal cells presented as a percentage of total cells? How are non-neuronal cells identified?

Please show the images corresponding to the analysis in Fig. 7. 

It is concerning that so many pups died in the control litters. Could this be a reflection of stress that is obscuring the study's findings?

Minor comments:

  1. There should be a space between the numbers and units, for example, n=5 should be n = 5. Please correct them throughout the manuscript.
  2. Every figure includes “Figur X” at the bottom of it; they are not necessary to present your results, so they should be removed.
  3. Table 2, the first row is empty, so it should be deleted.
  4. Figures 1 and 3 provide related results. If you applied a student, please clarify them in the corresponding figure legends.
  5. All your figure legends lack numbers.

Author Response (Reviewer 2): Practically all sections of the manuscript were rewritten entirely, experiments were redone, the number of experiments has been increased, and many reviewer suggestions were now included in that new version. The grammatical and typographical errors have been corrected and the text edited. The Introduction, Material, Method, and Discussion sections of the manuscript were revised, shortened, and completely rewritten to include the suggestions and comments of reviewers. 

All male or female animals after birth were used to measure birth weight.

Regarding the aspects raised by the reviewer about the animals used in the present study, it is appropriate to clarify that dams were housed in pairs under standard laboratory conditions and had access to food and water ad libitum.

This study aimed to evaluate the effects of Fluoxetine use during pregnancy, without involving experimental models of psychiatric diseases. The dosage of Fluoxetine used was consistent with well-established guidelines in the literature for rodents.

The Abstract, Introduction, Materials and Methods, and Discussion sections of the manuscript were revised and completely rewritten to include the suggestions and comments of Reviewer 1. In the present study, only male pups were studied and weighed on their birthday, and eight male puppies were kept per mother. On the 7th and 14th postnatal days, the brain, thymus, and adrenals from male NP and LP offspring were weighed. Here, it is crucial to note that sex hormones influence sexual phenotype dimorphism in the fetal-programmed disease model in adulthood through changes in the long-term regulation of neural, cardiac, and endocrine functions. Thus, the present study was performed in male rats, considering the findings above to eliminate interferences due to gender differences [Kwong et al., 2000; Gillette et al., 2017]. However, additional studies evaluating the functionality of the kidneys, lungs, and liver, as well as aspects related to the behavior of animals subjected to psychological or nutritional stresses of both genders, are being conducted in the laboratory. The results obtained will be discussed in the future.

The study demonstrated a decrease in the maternal care index among fluoxetine-treated dams during the initial postpartum days relative to the control group. While few studies have investigated postnatal maternal care following perinatal fluoxetine treatment, existing research presents conflicting results regarding the effects on maternal care behavior. In terms of anxiety-like behavior, the elevated plus maze test revealed no significant differences between offspring from control and fluoxetine-treated progeny, as indicated by similar entries in the open and closed arms, as correctly expressed throughout the abstract and main text. Previous studies are contradictory; while McAllister et al. (2012) and Kiryanova et al. described a reduced level of anxiety in gestational SSRI-treated offspring, others, however, did not show any changes. The current study's analysis revealed increased overall activity in the F-offspring compared to the control progeny, although average speed and total distance covered did not differ significantly between the groups. The F-treated offspring spent more time in the central area of the open field apparatus, suggesting reduced anxiety-related emotionality. However, the findings also indicated increased fecal bolus production, reflecting a heightened fear state in the F offspring. Previous studies by Kiryanova et al. examined the effects of prenatal stress in rodents treated with Fluoxetine, revealing hyperactivity in progeny exposed to stress without SSRI treatment. It is worth noting that Kiryanova et al. focused on time and distance traveled, which showed no change in the present study. Conversely, Noorlander et al. found that F offspring spent less time in the center of the open field than the control progeny, which contrasts with the current results. The Morris water maze study revealed a reduced capacity for learning and memory retention in F offspring following the removal of the hidden platform, compared to the control group. However, assessments of working memory, linked to the prefrontal cortex, and reference memory, associated with the hippocampus, did not show significant differences between the two groups. This indicates that the effects of selective serotonin reuptake inhibitor (SSRI) treatment on offspring behavior and memory retention are still not fully understood. Previous studies found no significant differences in learning and memory in rodents treated with SSRIs compared to controls, primarily because they did not use the hidden platform strategy for memory assessment. In this study, F progeny from SSRI-treated dams exhibited increased motor activity and fear behaviors in the open-field test, correlating with diminished learning and memory as measured by the Morris water maze. If confirmed through further research, these findings may greatly inform our understanding of SSRIs' effects on offspring behavior and memory retention, particularly concerning the treatment of attention deficit hyperactivity disorder (ADHD). To our knowledge, no prior research has linked SSRI usage to the development of ADHD in offspring rodents. Previous retrospective clinical studies have suggested an increased risk of autism spectrum disorders (ASD) in children exposed to antidepressants during the prenatal period. The prevalence of ASD and attention-deficit/hyperactivity disorder (ADHD) among children in the United States, particularly a combined incidence of nearly 30% in New England, underscores the need for further investigation. Multiple logistic regression analyses have shown an association between prenatal antidepressant exposure and ASD risk in 1,377 affected children and 2,243 diagnosed with ADHD. However, adjusting for socio-demographic factors revealed no significant relationship between antidepressant exposure during pregnancy and ASD risk. It is essential to consider the modest risk of ADHD and ASD resulting from prenatal exposure against the substantial consequences of untreated maternal depression. In another study by Figueroa (2010) involving approximately 38,000 children, fewer than 500 were diagnosed with ADHD, and there was no link between prior SSRI use and the condition, with a notable association with maternal psychiatric illness instead. Conversely, Clements (2015) found a direct association between gestational exposure to SSRI-class antidepressants and increased ADHD risk, excluding variations in maternal psychiatric illness. Furthermore, studies indicate that perinatal exposure to Fluoxetine modifies neural network architecture in key areas of the hippocampus, potentially leading to behavioral abnormalities. However, our current study, which utilized the low-cost, elevated plus maze (LCE) and open field tests, did not indicate increased anxiety-like behavior in offspring of dams treated with Fluoxetine during pregnancy and lactation, but did demonstrate significant changes in memory and learning in these offspring. Recent studies have revealed that the regulation of activity and exploration capacities in rodents is not solely influenced by postpartum maternal care. The interaction between hippocampal structural changes and reduced modulation of the prefrontal cortex—responsible for activity, exploration, and executive functions—along with the anterior cingulate gyrus and the amygdala, which are associated with attention, emotional responses, fear, and learning, may explain the findings of this research. Consequently, it can be hypothesized that behavioral changes are linked not only to modifications in maternal care but also to gestational exposure to selective serotonin reuptake inhibitors (SSRIs).

Cell and neuron quantification followed the methods discussed in sources [7,66]. Six-week-old Control (C) and age-matched Fluoxetine (F) offspring (n=5 per group) were anesthetized with isoflurane and perfused transcardially with saline and a 4% paraformaldehyde solution. The hippocampus was mechanically dissociated and homogenized in a solution of sodium citrate and Triton X-100. Nuclei were resuspended in a DAPI solution, and nuclear density was assessed using a hemocytometer. Total cell counts were estimated based on atomic density. Neurons were identified by incubating a portion of the nuclear suspension with an anti-NeuN antibody, followed by a Cy3-conjugated secondary antibody for counting under a fluorescence microscope.

Reviewer 3 Report

Comments and Suggestions for Authors

Methodological Aspects:The research is based on a comprehensive experimental design, with methodologies well-suited to the research questions posed. Multiple techniques were employed to ensure a multidimensional evaluation. At the behavioral level: Morris water maze, open field test, and elevated plus maze; at the physiological level: body weight, brain weight, and hippocampal mass; and at the histological level: immunohistochemistry and isotropic fractionation. Together, these approaches provide evidence from multiple levels of analysis and yield noteworthy findings.

Relevance of Findings:The observed increase in hippocampal neuron numbers in fluoxetine-exposed offspring, along with behavioral alterations, is an original and noteworthy finding. It raises new questions regarding pharmacologically induced neurogenesis. Furthermore, the study connects its findings with the “selfish brain” hypothesis, which proposes that the brain prioritizes its development even under adverse conditions, thereby adding an interesting theoretical dimension to the analysis.

Points to Be Addressed:

Construct and Predictive Validity: Although the techniques employed allow for comparison of results, it would be pertinent to clarify which of the behavioral or anatomical assessments have greater construct or predictive validity in relation to the study objectives.

Causal Interpretation: The discussion is generally well-constructed and integrates previous literature—both supportive and contradictory—to contextualize the findings. However, some paragraphs contain somewhat ambiguous and repetitive causal interpretations. The text suggests a “substantial association” between fluoxetine treatment, neurogenesis, and behavioral changes, but it remains unclear whether there is a direct causal link between increased neuronal counts and observed memory deficits or hyperactivity. Future studies might benefit from including additional manipulations, such as neurogenesis inhibitors, to clarify these relationships.

Behavioral Results Consistency: The simultaneous report of reduced anxiety (open field) and impaired learning and memory (Morris water maze) is noteworthy. However, no differences were found in working or reference memory. This may suggest that the alterations are not global but rather specific, which should be emphasized in the interpretation.

Suggestions for Tables and Figures:

Tables 1 and 2: These tables should be centered, and their explanatory captions should be placed below the table rather than above.

Figures: It is suggested to remove the “FIGURE 1, 2, 3,” etc., labels from beneath each image, since the figure legend already includes the figure number in its caption.

Figure 7A:The color of the bars does not match the reference color labels and should be corrected for visual accuracy and interpretation.

Comments on the Quality of English Language

It is recommended to revise grammatical and typographical errors. For instance, in the abstract, the phrase “reduction in parental care provided by fluoxetine-treated dams compared to control progeny” is confusing—it would be clearer to compare dams in the fluoxetine group to those in the control group. This may be a writing error that requires correction.

Author Response

Response from the Authors

Response to Reviewer Comments

International Journal of Molecular Science

Ref.: ijms-3693676

Title: Effect of Gestational Fluoxetine Administration on Behavioral Tests and Hippocampal Structure in Male Rat Offspring

Authors: Marcelo Gustavo Lopes, Gabriel Boer Grigoletti-Lima , Patricia Aline Boer , José Antonio Rocha Gontijo *

Comments from the Editors and Reviewers:

  1. Response to Reviewer #3

First, thank you very much for your criticism. We greatly appreciate your comments and suggestions. Practically all sections of the manuscript were rewritten entirely, experiments were redone, the number of experiments was increased, and many reviewer suggestions were now included in that new version. The grammatical and typographical errors have been corrected and the text edited. The Introduction, Material, Method, and Discussion sections of the manuscript were revised, shortened, and completely rewritten to include the suggestions and comments of reviewers. 

Reviewer #3: Methodological Aspects: The research is based on a comprehensive experimental design, with methodologies well-suited to the research questions posed. Multiple techniques were employed to ensure a multidimensional evaluation. At the behavioral level: Morris water maze, open field test, and elevated plus maze; at the physiological level: body weight, brain weight, and hippocampal mass; and at the histological level: immunohistochemistry and isotropic fractionation. Together, these approaches provide evidence from multiple levels of analysis, yielding noteworthy findings.

Relevance of Findings: The observed increase in hippocampal neuron numbers in fluoxetine-exposed offspring, along with behavioral alterations, is an original and noteworthy finding. It raises new questions regarding pharmacologically induced neurogenesis. Furthermore, the study connects its findings with the "selfish brain" hypothesis, which proposes that the brain prioritizes its development even under adverse conditions, thereby adding an interesting theoretical dimension to the analysis.

Points to Be Addressed:

Construct and Predictive Validity: Although the techniques employed allow for comparison of results, it would be pertinent to clarify which of the behavioral or anatomical assessments have greater construct or predictive validity in relation to the study objectives.

Causal Interpretation: The discussion is generally well-constructed and integrates previous literature—both supportive and contradictory—to contextualize the findings. However, some paragraphs contain somewhat ambiguous and repetitive causal interpretations. The text suggests a "substantial association" between n fluoxetine treatment, neurogenesis, and behavioral changes, but it remains unclear whether there is a direct causal link between increased neuronal counts and observed memory deficits or hyperactivity. Future studies might benefit from including additional manipulations, such as neurogenesis inhibitors, to clarify these relationships.

Behavioral Results Consistency: The simultaneous report of reduced anxiety (open field) and impaired learning and memory (Morris water maze) is noteworthy. However, no differences were found in working or reference memory. This suggests that the alterations may not be global, but rather specific, which should be emphasized in the interpretation.

Suggestions for Tables and Figures:

Tables 1 and 2: These tables should be centered, and their explanatory captions should be placed below the table rather than above.

Figures: It is suggested to remove the "FIGURE 1, 2, 3," etc., labels from beneath each image, since the figure legend already includes the figure number in its caption.

Figure 7A: The color of the bars does not match the reference color labels and should be corrected for visual accuracy and interpretation.

Comments on the Quality of the English Language

It is recommended to revise grammatical and typographical errors. For instance, in the abstract, the phrase "reduct" in parental care provided by fluoxetine-treated dams compared to control progeny" is confusing—it would be clearer to compare dams in the fluoxetine group to those in the control group. This may be a writing error that requires correction.

Author Response (Reviewer 3): Practically all sections of the manuscript were rewritten entirely, experiments were redone, the number of experiments has been increased, and many reviewer suggestions were now included in that new version. The grammatical and typographical errors have been corrected and the text edited. The Introduction, Material, Method, and Discussion sections of the manuscript were revised, shortened, and completely rewritten to include the suggestions and comments of reviewers. 

All male or female animals after birth were used to measure birth weight.

Regarding the aspects raised by the reviewer about the animals used in the present study, it is appropriate to clarify that dams were housed in pairs under standard laboratory conditions and had access to food and water ad libitum.

This study aimed to evaluate the effects of Fluoxetine use during pregnancy, without involving experimental models of psychiatric diseases. The dosage of Fluoxetine used was consistent with well-established guidelines in the literature for rodents.

The Abstract, Introduction, Materials and Methods, and Discussion sections of the manuscript were revised and completely rewritten to include the suggestions and comments of Reviewer 1. In the present study, only male pups were studied and weighed on their birthday, and eight male puppies were kept per mother. On the 7th and 14th postnatal days, the brain, thymus, and adrenals from male NP and LP offspring were weighed. Here, it is crucial to note that sex hormones influence sexual phenotype dimorphism in the fetal-programmed disease model in adulthood through changes in the long-term regulation of neural, cardiac, and endocrine functions. Thus, the present study was performed in male rats, considering the findings above to eliminate interferences due to gender differences [Kwong et al., 2000; Gillette et al., 2017]. However, additional studies evaluating the functionality of the kidneys, lungs, and liver, as well as aspects related to the behavior of animals subjected to psychological or nutritional stresses of both genders, are being conducted in the laboratory. The results obtained will be discussed in the future.

The study demonstrated a decrease in the maternal care index among fluoxetine-treated dams during the initial postpartum days relative to the control group. While few studies have investigated postnatal maternal care following perinatal fluoxetine treatment, existing research presents conflicting results regarding the effects on maternal care behavior. In terms of anxiety-like behavior, the elevated plus maze test revealed no significant differences between offspring from control and fluoxetine-treated progeny, as indicated by similar entries in the open and closed arms, as correctly expressed throughout the abstract and main text. Previous studies are contradictory; while McAllister et al. (2012) and Kiryanova et al. described a reduced level of anxiety in gestational SSRI-treated offspring, others, however, did not show any changes. The current study's field analysis revealed increased overall activity in the F-offspring compared to the control progeny, although average speed and total distance covered did not differ significantly between the groups. The F-treated offspring spent more time in the central area of the open field apparatus, suggesting reduced anxiety-related emotionality. However, the findings also indicated increased fecal bolus production, reflecting a heightened fear state in the F offspring. Previous studies by Kiryanova et al. examined the effects of prenatal stress in rodents treated with Fluoxetine, revealing hyperactivity in progeny exposed to stress without SSRI treatment. It is worth noting that Kiryanova et al. focused on time and distance traveled, which showed no change in the present study. Conversely, Noorlander et al. found that F offspring spent less time in the center of the open field than the control progeny, which contrasts with the current results. The Morris water maze study revealed a reduced capacity for learning and memory retention in F offspring following the removal of the hidden platform, compared to the control group. However, assessments of working memory, linked to the prefrontal cortex, and reference memory, associated with the hippocampus, did not show significant differences between the two groups. This indicates that the effects of selective serotonin reuptake inhibitor (SSRI) treatment on offspring behavior and memory retention are still not fully understood. Previous studies found no significant differences in learning and memory in rodents treated with SSRIs compared to controls, primarily because they did not use the hidden platform strategy for memory assessment. In this study, F progeny from SSRI-treated dams exhibited increased motor activity and fear behaviors in the open-field test, correlating with diminished learning and memory as measured by the Morris water maze. If confirmed through further research, these findings may greatly inform our understanding of SSRIs' effects on offspring behavior and memory retention, particularly concerning the treatment of attention deficit hyperactivity disorder (ADHD). To our knowledge, no prior research has linked SSRI usage to the development of ADHD in offspring rodents. Previous retrospective clinical studies have suggested an increased risk of autism spectrum disorders (ASD) in children exposed to antidepressants during the prenatal period. The prevalence of ASD and attention-deficit/hyperactivity disorder (ADHD) among children in the United States, particularly a combined incidence of nearly 30% in New England, underscores the need for further investigation. Multiple logistic regression analyses have shown an association between prenatal antidepressant exposure and ASD risk in 1,377 affected children and 2,243 diagnosed with ADHD. However, adjusting for socio-demographic factors revealed no significant relationship between antidepressant exposure during pregnancy and ASD risk. It is essential to consider the modest risk of ADHD and ASD resulting from prenatal exposure against the substantial consequences of untreated maternal depression. In another study by Figueroa (2010) involving approximately 38,000 children, fewer than 500 were diagnosed with ADHD, and there was no link between prior SSRI use and the condition, with a notable association with maternal psychiatric illness instead. Conversely, Clements (2015) found a direct association between gestational exposure to SSRI-class antidepressants and increased ADHD risk, excluding variations in maternal psychiatric illness. Furthermore, studies indicate that perinatal exposure to Fluoxetine modifies neural network architecture in key areas of the hippocampus, potentially leading to behavioral abnormalities. However, our current study, which utilized the low-cost, elevated plus maze (LCE) and open field tests, did not indicate increased anxiety-like behavior in offspring of dams treated with Fluoxetine during pregnancy and lactation but did demonstrate significant changes in memory and learning in these offspring. Recent studies have revealed that the regulation of activity and exploration capacities in rodents is not solely influenced by postpartum maternal care. The interaction between hippocampal structural changes and reduced modulation of the prefrontal cortex—responsible for activity, exploration, and executive functions—along with the anterior cingulate gyrus and the amygdala, which are associated with attention, emotional responses, fear, and learning, may explain the findings of this research. Consequently, it can be hypothesized that behavioral changes are linked not only to modifications in maternal care but also to gestational exposure to selective serotonin reuptake inhibitors (SSRIs).

Cell and neuron quantification followed the methods discussed in sources [7,66]. Six-week-old Control (C) and age-matched Fluoxetine (F) offspring (n=5 per group) were anesthetized with isoflurane and perfused transcardially with saline and a 4% paraformaldehyde solution. The hippocampus was mechanically dissociated and homogenized in a solution of sodium citrate and Triton X-100. Nuclei were resuspended in a DAPI solution, and nuclear density was assessed using a hemocytometer. Total cell counts were estimated based on atomic density. Neurons were identified by incubating a portion of the nuclear suspension with an anti-NeuN antibody, followed by a Cy3-conjugated secondary antibody for counting under a fluorescence microscope.

Reviewer 4 Report

Comments and Suggestions for Authors

I appreciate the opportunity to review an interesting and apparently well-crafted work. However, you should anticipate that I believe the characteristics of this work are not appropriate for full-text journal publication. Its circumstantial, experimental characteristics, and a certain lack of correspondence between stated objectives and results, lead me to believe that it is suitable for publication in its current format, unless several aspects are thoroughly reviewed and improved.
The manuscript evaluates the effects of intrauterine and lactational exposure to fluoxetine on behavior and memory using an animal model that has not been fully validated.
This is a very interesting manuscript that addresses an important topic with clinical application using an animal model. Even so, it is not a significant contribution, given that there is clinical evidence (reports and cohort studies) that have provided better results. It provides little real value for clinical application. It does not change the knowledge on the subject. I consider the work more appropriate for a short presentation or abstract at a scientific meeting than for a journal manuscript (at least in its current format).
Specifically analyzing the sections: The title is appropriate, brief, concise, and adequately descriptive.
The abstract is confusing, not divided into sections, lacks specific numerical results, and mixes conclusions with general statements. It should be rewritten, with an appropriate structure and providing numerical results and statistical parameters.
The introduction is too long and should be shortened. It presents a presumed justification for the study very well, and the research objectives are adequately presented. However, it is worth mentioning that the results described subsequently do not match the stated objectives.
The materials and methods are very well described. A methodology is used that appears appropriate and reproducible. The description is complete and detailed, ensuring the reproducibility of the study. However, methodological validation seems lacking, especially the correlation studied with respect to possible implications in humans.
The results are interesting and well presented. Graphic tools are used to improve the presentation. No significant biases were detected. The discussion is acceptable, comprehensive, and considers various aspects of comparison with other studies. However, there is a significant shortcoming in the description of the limitations of this work. The latter should be explained in more detail: the use of an animal model, which is not fully applicable to humans, and the need for specific clinical studies.
The figures are of acceptable quality but are poorly designed, including large-print captions with the figure number and excessive white space in the graph.
The conclusions are vague and insignificant, and unrelated to the study's objective. They show a reduction in body weight and no effect on the hippocampus. Furthermore, it proposes results regarding parental control that were not adequately targeted.
The references are excessive for an original work that does not adequately support the studies conducted.

Author Response

Response from the Authors

Response to Reviewer Comments

International Journal of Molecular Science

Ref.: ijms-3693676

Title: Effect of Gestational Fluoxetine Administration on Behavioral Tests and Hippocampal Structure in Male Rat Offspring

Authors: Marcelo Gustavo Lopes, Gabriel Boer Grigoletti-Lima, Patricia Aline Boer , José Antonio Rocha Gontijo *

Comments from the Editors and Reviewers:

  1. Response to Reviewer #4

First, thank you very much for your criticism. We greatly appreciate your comments and suggestions. Practically all sections of the manuscript were rewritten entirely, experiments were repeated, the number of experiments was increased, and many reviewer suggestions were incorporated into the new version. The grammatical and typographical errors have been corrected and the text edited. The Introduction, Material, Method, and Discussion sections of the manuscript were revised, shortened, and completely rewritten to include the suggestions and comments of reviewers. 

Reviewer #4: I appreciate the opportunity to review an engaging and well-crafted work. However, you should anticipate that I believe the characteristics of this work are not appropriate for full-text journal publication. Its circumstantial, experimental characteristics, and a certain lack of correspondence between stated objectives and results, lead me to believe that it is suitable for publication in its current format, unless several aspects are thoroughly reviewed and improved.
The manuscript evaluates the effects of intrauterine and lactational exposure to Fluoxetine on behavior and memory using an animal model that has not been fully validated.
This is a very interesting manuscript that addresses an important topic with clinical application using an animal model. Even so, it is not a significant contribution, given that there is clinical evidence (reports and cohort studies) that have provided better results. It provides little real value for clinical application. It does not change the knowledge on the subject. I consider the work more appropriate for a short presentation or abstract at a scientific meeting than for a journal manuscript (at least in its current format).
Specifically analyzing the sections: The title is appropriate, brief, concise, and adequately descriptive.
The abstract is confusing, lacks clear sections, and fails to present specific numerical results, instead mixing conclusions with general statements. It should be rewritten with an appropriate structure, providing numerical results and statistical parameters.
The introduction is too long and should be shortened. It presents a presumed justification for the study very well, and the research objectives are adequately presented. However, it is worth mentioning that the results described subsequently do not match the stated objectives.
The materials and methods are very well described. A methodology is used that appears appropriate and reproducible. The description is complete and detailed, ensuring the reproducibility of the study. However, methodological validation seems lacking, especially the correlation studied with respect to possible implications in humans.
The results are interesting and well presented. Graphic tools are used to improve the presentation. No significant biases were detected. The discussion is acceptable, comprehensive, and considers various aspects of comparison with other studies. However, there is a significant shortcoming in the description of the limitations of this work. The latter should be explained in more detail: the use of an animal model, which is not fully applicable to humans, and the need for specific clinical studies.
The figures are of acceptable quality but are poorly designed, including large-print captions with the figure number and excessive white space in the graph.
The conclusions are vague and insignificant, and unrelated to the study's objective. They show a reduction in body weight and no effect on the hippocampus. Furthermore, it proposes results regarding parental control that were not adequately targeted.
The references are excessive for an original work that does not adequately support the studies conducted.

Author Response (Reviewer 4): Practically all sections of the manuscript were rewritten entirely, experiments were redone, the number of experiments has been increased, and many reviewer suggestions were now included in that new version. The grammatical and typographical errors have been corrected and the text edited. The Introduction, Material, Method, and Discussion sections of the manuscript were revised, shortened, and completely rewritten to include the suggestions and comments of reviewers. 

All male or female animals after birth were used to measure birth weight.

Regarding the aspects raised by the reviewer about the animals used in the present study, it is appropriate to clarify that dams were housed in pairs under standard laboratory conditions and had access to food and water ad libitum.

This study aimed to evaluate the effects of Fluoxetine use during pregnancy, without involving experimental models of psychiatric diseases. The dosage of Fluoxetine used was consistent with well-established guidelines in the literature for rodents.

The Abstract, Introduction, Materials and Methods, and Discussion sections of the manuscript were revised and completely rewritten to include the suggestions and comments of Reviewer 1. In the present study, only male pups were studied and weighed on their birthday, and eight male puppies were kept per mother. On the 7th and 14th postnatal days, the brain, thymus, and adrenals from male NP and LP offspring were weighed. Here, it is crucial to note that sex hormones influence sexual phenotype dimorphism in the fetal-programmed disease model in adulthood through changes in the long-term regulation of neural, cardiac, and endocrine functions. Thus, the present study was performed in male rats, considering the findings above to eliminate interferences due to gender differences [Kwong et al., 2000; Gillette et al., 2017]. However, additional studies evaluating the functionality of the kidneys, lungs, and liver, as well as aspects related to the behavior of animals subjected to psychological or nutritional stresses of both genders, are being conducted in the laboratory. The results obtained will be discussed in the future.

The study demonstrated a decrease in the maternal care index among fluoxetine-treated dams during the initial postpartum days relative to the control group. While few studies have investigated postnatal maternal care following perinatal fluoxetine treatment, existing research presents conflicting results regarding the effects on maternal care behavior. In terms of anxiety-like behavior, the elevated plus maze test revealed no significant differences between offspring from control and fluoxetine-treated progeny, as indicated by similar entries in the open and closed arms, as correctly expressed throughout the abstract and main text. Previous studies are contradictory; while McAllister et al. (2012) and Kiryanova et al. described a reduced level of anxiety in gestational SSRI-treated offspring, others, however, did not show any changes. The current study's open-field analysis revealed increased overall activity in the F-offspring compared to the control progeny, although average speed and total distance covered did not differ significantly between the groups. The F-treated offspring spent more time in the central area of the open field apparatus, suggesting reduced anxiety-related emotionality. However, the findings also indicated increased fecal bolus production, reflecting a heightened fear state in the F offspring. Previous studies by Kiryanova et al. examined the effects of prenatal stress in rodents treated with Fluoxetine, revealing hyperactivity in progeny exposed to stress without SSRI treatment. It is worth noting that Kiryanova et al. focused on time and distance traveled, which showed no change in the present study. Conversely, Noorlander et al. found that F offspring spent less time in the center of the open field than the control progeny, which contrasts with the current results. The Morris water maze study revealed a reduced capacity for learning and memory retention in F offspring following the removal of the hidden platform, compared to the control group. However, assessments of working memory, linked to the prefrontal cortex, and reference memory, associated with the hippocampus, did not show significant differences between the two groups. This indicates that the effects of selective serotonin reuptake inhibitor (SSRI) treatment on offspring behavior and memory retention are still not fully understood. Previous studies found no significant differences in learning and memory in rodents treated with SSRIs compared to controls, primarily because they did not use the hidden platform strategy for memory assessment. In this study, F progeny from SSRI-treated dams exhibited increased motor activity and fear behaviors in the open-field test, correlating with diminished learning and memory as measured by the Morris water maze. If confirmed through further research, these findings may greatly inform our understanding of SSRIs' impact on offspring behavior and memory retention, particularly concerning the treatment of attention deficit hyperactivity disorder (ADHD). To our knowledge, no prior research has linked SSRI usage to the development of ADHD in offspring rodents. Previous retrospective clinical studies have suggested an increased risk of autism spectrum disorders (ASD) in children exposed to antidepressants during the prenatal period. The prevalence of ASD and attention-deficit/hyperactivity disorder (ADHD) among children in the United States, particularly a combined incidence of nearly 30% in New England, underscores the need for further investigation. Multiple logistic regression analyses have shown an association between prenatal antidepressant exposure and ASD risk in 1,377 affected children and 2,243 diagnosed with ADHD. However, adjusting for socio-demographic factors revealed no significant relationship between antidepressant exposure during pregnancy and ASD risk. It is essential to consider the modest risk of ADHD and ASD resulting from prenatal exposure against the substantial consequences of untreated maternal depression. In another study by Figueroa (2010) involving approximately 38,000 children, fewer than 500 were diagnosed with ADHD, and there was no link between prior SSRI use and the condition, with a notable association with maternal psychiatric illness instead. Conversely, Clements (2015) found a direct association between gestational exposure to SSRI-class antidepressants and increased ADHD risk, excluding variations in maternal psychiatric illness. Furthermore, studies indicate that perinatal exposure to Fluoxetine modifies neural network architecture in key areas of the hippocampus, potentially leading to behavioral abnormalities. However, our current study, which utilized the low-cost, elevated plus maze (LCE) and open field tests, did not indicate increased anxiety-like behavior in offspring of dams treated with Fluoxetine during pregnancy and lactation, but did demonstrate significant changes in memory and learning in these offspring. Recent studies have revealed that the regulation of activity and exploration capacities in rodents is not solely influenced by postpartum maternal care. The interaction between hippocampal structural changes and reduced modulation of the prefrontal cortex—responsible for activity, exploration, and executive functions—along with the anterior cingulate gyrus and the amygdala, which are associated with attention, emotional responses, fear, and learning, may explain the findings of this research. Consequently, it can be hypothesized that behavioral changes are linked not only to modifications in maternal care but also to gestational exposure to selective serotonin reuptake inhibitors (SSRIs).

Cell and neuron quantification followed the methods discussed in sources [7, 66]. Six-week-old Control (C) and age-matched Fluoxetine (F) offspring (n=5 per group) were anesthetized with isoflurane and perfused transcardially with saline and a 4% paraformaldehyde solution. The hippocampus was mechanically dissociated and homogenized in a solution of sodium citrate and Triton X-100. Nuclei were resuspended in a DAPI solution, and nuclear density was assessed using a hemocytometer. Total cell counts were estimated based on atomic density. Neurons were identified by incubating a portion of the nuclear suspension with an anti-NeuN antibody, followed by a Cy3-conjugated secondary antibody for counting under a fluorescence microscope.

Round 2

Reviewer 2 Report

Comments and Suggestions for Authors

slightly improved!

Author Response

Response from the Authors

Response to Reviewer Comments

International Journal of Molecular Science

Ref.: ijms-3693676

Title: Effect of Gestational Fluoxetine Administration on Behavioral Tests and Hippocampal Structure in Male Rat Offspring

Authors: Marcelo Gustavo Lopes, Gabriel Boer Grigoletti-Lima , Patricia Aline Boer , José Antonio Rocha Gontijo *

Comments from the Editors and Reviewers:

  1. Response to Reviewer #2

At first, thank you very much for your criticisms. We greatly appreciate your comments and suggestions. Practically all sections of the manuscript were rewritten entirely, experiments were redone, the number of experiments has been increased, and many reviewer suggestions were now included in that new version. The grammatical and typographical errors have been corrected and the text edited. The Introduction, Material, Method, and Discussion sections of the manuscript were revised, shortened, and completely rewritten to include the suggestions and comments of reviewers. 

Reviewer #2: This research examines behavioral and neuroanatomical alterations following offspring' exposure to fluoxetine. However, one of my major concerns with this work is that it does not address whether the observed changes are due to the fluoxetine exposure or changes in maternal care, which could be done by rearing control pups with a fluoxetine mother and vice versa. 

Additionally, the methods and title suggest that the experiments are done in male offspring only, but then Fig. 1A shows females. Could the others please clarify whether all other experiments were performed with males only or both males and females?

The abstract states that they found a reduction in anxiety-like behaviors in male offspring but then there is no supporting evidence for this since there are no changes in the elevated plus maze.

It is unclear how the analysis in Figure 6 was conducted. Why aren't neurons and non-neuronal cells presented as a percentage of total cells? How are non-neuronal cells identified?

Please show the images corresponding to the analysis in Fig. 7. 

It is concerning that so many pups died in the control litters. Could this be a reflection of stress that is obscuring the study's findings?

Minor comments:

  1. There should be a space between the numbers and units, for example, n=5 should be n = 5. Please correct them throughout the manuscript.
  2. Every figure includes “Figure X” in the bottom of them, they are not necessary to present your results, so they should be removed.
  3. Table 2, the first row is empty, so it should be deleted.
  4. Figure 1, as well as Figure 3, for related results, if you applied student’s t-test, you should clarify them in the corresponding figure legends.
  5. All your figure legends are lack of n numbers.

Author Response (Reviewer 2): Practically all sections of the manuscript were rewritten entirely, experiments were redone, the number of experiments has been increased, and many reviewer suggestions were now included in that new version. The grammatical and typographical errors have been corrected and the text edited. The Introduction, Material, Method, and Discussion sections of the manuscript were revised, shortened, and completely rewritten to include the suggestions and comments of reviewers. 

All male or female animals after birth were used to measure birth weight.

Regarding the aspects raised by the reviewer about the animals used in the present study, it is appropriate to clarify that dams were housed in pairs under standard laboratory conditions and had access to food and water ad libitum.

This study aimed to evaluate the effects of Fluoxetine use during pregnancy, without involving experimental models of psychiatric diseases. The dosage of Fluoxetine used was consistent with well-established guidelines in the literature for rodents.

The Abstract, Introduction, Material, Method, and Discussion sections of the manuscript were revised and completely rewritten to include the suggestions and comments of reviewer 1. In the present study, all male or female animals after birth were used to measure birth weight, however, only the male pups were studied. Here, it is critical to state that sex hormones determine sexual phenotype dimorphism in the fetal-programmed disease model in adulthood by changes in the long-term control of neural, cardiac, and endocrine functions. Thus, the present study was performed in male rats, considering the findings above to eliminate interferences due to gender differences [Kwong et al., 2000; Gillette et al., 2017]. However, additional studies evaluating the functionality of the kidneys, lungs, and liver and aspects related to the behavior of animals subjected to psychological or nutritional stresses of both genders are being conducted in the laboratory, and the results obtained will be discussed in the future.

The study demonstrated a decrease in the maternal care index among fluoxetine-treated dams during the initial postpartum days relative to the control group. While few studies have investigated postnatal maternal care following perinatal fluoxetine treatment, existing research presents conflicting results regarding the effects on maternal care behavior. In terms of anxiety-like behavior, the elevated plus maze test revealed no significant differences between offspring from control and fluoxetine-treated progeny, as indicated by similar entries in the open and closed arms, as now correctly express throughout the abstract as well as the main text. Previous studies are contradictory; while McAllister et al. (2012) and Kiryanova et al. described a reduced level of anxiety in gestational SSRI-treated offspring, others however, did not show any changes. The current study's open-field analysis revealed increased overall activity in the F-offspring compared to the control progeny, although average speed and total distance covered did not differ significantly between the groups. The F-treated offspring spent more time in the central area of the open field apparatus, suggesting reduced anxiety-related emotionality. However, the findings also indicated increased fecal bolus production, reflecting a heightened fear state in the F offspring. Previous studies by Kiryanova et al. examined the effects of prenatal stress in rodents treated with fluoxetine, revealing hyperactivity in progeny exposed to stress without SSRI treatment. It is worth noting that Kiryanova et al. focused on time and distance traveled, which showed no change in the present study. Conversely, Noorlander et al. found that F offspring spent less time in the center of the open field than the control progeny, which contrasts with the current results. The Morris water maze study revealed a reduced capacity for learning and memory retention in F offspring following the removal of the hidden platform, compared to the control group. However, assessments of working memory, linked to the prefrontal cortex, and reference memory, associated with the hippocampus, did not show significant differences between the two groups. This indicates that the effects of selective serotonin reuptake inhibitor (SSRI) treatment on offspring behavior and memory retention are still not fully understood. Previous studies found no significant differences in learning and memory in rodents treated with SSRIs compared to controls, primarily because they did not use the hidden platform strategy for memory assessment. In this study, F progeny from SSRI-treated dams exhibited increased motor activity and fear behaviors in the open-field test, correlating with diminished learning and memory as measured by the Morris water maze. If confirmed through further research, these findings may greatly inform our understanding of SSRIs' impact on offspring behavior and memory retention, particularly concerning the treatment of attention deficit hyperactivity disorder (ADHD). To our knowledge, no prior research has linked SSRI usage to the development of ADHD in offspring rodents. Previous retrospective clinical studies have suggested an increased risk of autism spectrum disorders (ASD) in children exposed to antidepressants during the prenatal period. The prevalence of ASD and attention-deficit/hyperactivity disorder (ADHD) among children in the United States, particularly a combined incidence of nearly 30% in New England, underscores the need for further investigation. In multiple logistic regression analyses have shown an association between prenatal antidepressant exposure and ASD risk in 1,377 affected children and 2,243 diagnosed with ADHD. However, adjusting for socio-demographic factors revealed no significant relationship between antidepressant exposure during pregnancy and ASD risk. It is essential to consider the modest risk of ADHD and ASD resulting from prenatal exposure against the substantial consequences of untreated maternal depression. In another study by Figueroa (2010) involving approximately 38,000 children, fewer than 500 were diagnosed with ADHD, and there was no link between prior SSRI use and the condition, with a notable association with maternal psychiatric illness instead. Conversely, Clements (2015) found a direct association between gestational exposure to SSRI-class antidepressants and increased ADHD risk, excluding variations in maternal psychiatric illness. Furthermore, studies indicate that perinatal exposure to fluoxetine modifies neural network architecture in key areas of the hippocampus, potentially leading to behavioral abnormalities. However, our current study, which utilized the low-cost, elevated plus maze (LCE) and open field tests, did not indicate increased anxiety-like behavior in offspring of dams treated with fluoxetine during pregnancy and lactation but did demonstrate significant changes in memory and learning in these offspring. Recent studies have revealed that the regulation of activity and exploration capacities in rodents is not solely influenced by postpartum maternal care. The interaction between hippocampal structural changes and reduced modulation of the prefrontal cortex—responsible for activity, exploration, and executive functions—along with the anterior cingulate gyrus and the amygdala, which are associated with attention, emotional responses, fear, and learning, may explain the findings of this research. Consequently, it can be hypothesized that behavioral changes are linked not only to modifications in maternal care but also to gestational exposure to selective serotonin reuptake inhibitors (SSRIs).

Cell and neuron quantification followed methods discussed in sources [7,66]. Six-week-old Control (C) and age-matched Fluoxetine (F) offspring (n=5 per group) were anesthetized with isoflurane and perfused transcardially with saline and a 4% paraformaldehyde solution. The hippocampus was mechanically dissociated and homogenized in a solution of sodium citrate and Triton X-100. Nuclei were resuspended in a DAPI solution, and nuclear density was assessed using a hemocytometer. Total cell counts were estimated based on nuclear density. Neurons were identified by incubating a portion of the nuclear suspension with an anti-NeuN antibody, followed by a Cy3-conjugated secondary antibody for counting under a fluorescence microscope.